# Diagnostic accuracy of blood tests of inflammation in paediatric appendicitis: a systematic review and meta-analysis

David Fawkner-Corbett [1,2,3] Gail Hayward [1] Mohammed Alkhmees [4]
Ann Van Den Bruel [5] Jose M Ordóñez-Mena [1,6] Gea A Holtman [1,4]

For numbered affiliations see end of article.

**Correspondence to**
Dr David Fawkner-Corbett;
david.fawkner-corbett@imm.ox.ac.uk

## ABSTRACT

**Objective** Possible childhood appendicitis is a common emergency presentation. The exact value of blood tests is debated. This study sought to determine the diagnostic accuracy of four blood tests (white cell count (WCC), neutrophil(count or percentage), C reactive protein (CRP) and/or procalcitonin) for childhood appendicitis.

**Design** A systematic review and diagnostic meta-analysis. Data sources included MEDLINE, EMBASE, Central, Web of Science searched from inception-March 2022 with reference searching and authors contacted for missing/unclear data. Eligibility criteria was studies reporting the diagnostic accuracy of the four blood tests compared to the reference standard (histology or follow-up). Risk of bias was assessed (QUADAS-2), pooled sensitivity and specificity were generated for each test and commonly presented cut-offs. To provide insight into clinical impact, we present strategies using a hypothetical cohort.

**Results** 67 studies were included (34 839 children, 13 342 with appendicitis), all in the hospital setting. The most sensitive tests were WCC (≥10 000 cells/µL, 53 studies sensitivity 0.85 (95% CI 0.80 to 0.89)) and absolute neutrophil count (ANC) (≥7500 cells/µL, five studies sensitivity 0.90 (95% CI 0.85 to 0.94)). Combination of WCC or CRP increased sensitivity further(≥10 000 cells/µL or ≥10 mg/L, individual patient data (IPD) of 6 studies, 0.97 (95% CI 0.93 to 0.99)). Applying results to a hypothetical cohort(1000 children with appendicitis symptoms, of whom 400 have appendicitis) 60 and 40 children would be wrongly discharged based solely on WCC and ANC, respectively, 12 with combination of WCC or CRP.

The most specific tests were CRP alone (≥50 mg/L, 38 studies, specificity 0.87 (95% CI 0.80 to 0.91)) or combined with WCC (≥10 000 cells/µL and ≥50 mg/L, IPD of six studies, 0.93 (95% CI 0.91 to 0.95)).

**Conclusions** The best performing single blood tests for ruling-out paediatric appendicitis are WCC or ANC; with accuracy improved combining WCC and CRP. These tests could be used at the point of care in combination with clinical prediction rules. We provide insight into the best cut-offs for clinical application.

**PROSPERO registration number** CRD42017080036

## STRENGTHS AND LIMITATIONS OF THIS STUDY

⇒ This systematic review and meta-analysis looked at the accuracy of the four most commonly used blood tests for diagnosing appendicitis against a reference standard of histology/surgical diagnosis or follow-up.

⇒ An extensive literature search for all studies reporting children presenting with symptoms of appendicitis reporting metrics for white cell count, neutrophil (count or percentage), C reactive protein and/or procalcitonin were included.

⇒ Authors were contacted when data was unclear, generating a pool of individual patient data to allow use of new methods and information about the cut offs that perform best for clinical application for all tests.

⇒ The literature is limited for application of results in primary care or as point of care tests, as no studies reported diagnostic accuracy in these settings.

⇒ Results were applied to a hypothetical cohort to aid clinical interpretation.

(ED).[1] Differentiation between self-limiting and surgical conditions, such as acute appendicitis, can be especially difficult in children where less than 50% of patients present with classically described symptoms.[2–4] Acute appendicitis is the most common surgical condition with over 12 000 children undergoing emergency surgery for the condition each year in England.[5] It is vital to prevent an unnecessary operation in children, which is reported in 5%–20% of cases,[6–9] while also undertaking prompt treatment and avoiding inappropriate discharge to mitigate the risks of more severe advanced disease including perforation.[10 11] Radiological investigations including CT imaging, MRI or ultrasound are useful adjunct to decide who needs surgery.[7 12] However, these modalities will often require admission and routine use in all suspected cases would increase the burden on the healthcare system and children.[13]

## INTRODUCTION

Abdominal pain is a common reason for a child to attend the emergency department

Blood tests are frequently used to investigate children with suspected appendicitis to help inform diagnostic decisions; with white cell count (WCC), neutrophils (as count or percentage) and C reactive protein (CRP) being most widely reported.[14][15] Procalcitonin is also increasingly used to distinguish paediatric sepsis.[16] Much of the evidence for their accuracy in this setting is applied from adult studies.[17][18] All these blood tests can be available within hours and are also increasingly available rapidly as point-of-care (POC) tests thus extending their future application to expedite the clinical decision-making process and broaden their use into community care.[19]

Clinical scoring tools, which can incorporate clinical signs and blood results, can be used to triage patients at hospital to determine the need for imaging or surgery.[20] Although the clinical prediction scores are recommended their reported accuracy and uptake varies widely, and use is more limited in children.[21] In addition, a core scoring criteria of each is blood tests with the type of test and cut-off varying, reflecting the fact the evidence of optimum diagnostic cut-off is scarce.[13][22]

Taken together, blood tests are widely performed but exact accuracy for informing the diagnosis of paediatric appendicitis is debated.[23] More insight in the diagnostic accuracy of several cut offs of blood tests would help to choose which POC tests should be prioritised for implementation in the ED, which should best inform admission or discharge decisions, and which cut-offs are optimum to integrate with future optimisation of clinical scoring tools. To date, a large, systematic appraisal of the diagnostic accuracy of these tests in paediatric appendicitis is lacking and so this study aims to undertake a systematic review and meta-analysis of all studies reporting the diagnostic accuracy of common blood tests in the diagnosis of paediatric appendicitis.

## METHODS

This systematic review and meta-analysis was undertaken in line with the Preferred Reporting Items for Systematic Reviews and Meta-Analyses guidelines[24] and was prospectively registered (PROSPERO: CRD42017080036).

### Public and patient involvement

The concept of this review involved discussion with patients and families before it was undertaken and discussed experience in paediatric blood tests, identifying it as important to know the exact value (precision antimicrobial prescribing patient and public involvement group, University of Oxford). The results of this manuscript were also presented in this forum which helped formulate the write up and key discussion points within the final manuscript.

### Search strategy

Searching of databases (Cochrane Database of Systematic Reviews, Cochrane Central Register of controlled trials, Database of abstracts of reviews of effects(DARE), Embase, Ovid(Medline) Epub Ahead of print in process and other Non-Indexed Citations, Ovid Medline(R) Daily, Ovid Medline(R) and Science Citation Index and Conference Proceedings Citation Index (SCIENCE)) included keyword combinations to identify children with symptoms suggestive of appendicitis having one of the four blood tests (WCC, neutrophil, CRP and procalcitonin) for diagnosis. Searching was performed on 15 March 2022 to include all relevant studies from database inception with no language or date limitations. Full search strategy is provided in online supplemental material 1. Handsearching of included studies references was undertaken. Assessment of foreign language studies was completed by a native language speaker.

### Study selection and data extraction

Included studies reported children (<18 years) presenting with symptoms suggestive of appendicitis that would have testing of WCC, neutrophil, CRP and/or procalcitonin, by laboratory or POC, to distinguish appendicitis from non-surgical cause.

For studies that were included the optimal reference standard for confirming diagnosis of appendicitis was decided as histology after surgery (ideal standard) or; imaging confirming appendicitis in non-operatively managed patients or surgical diagnosis in studies without histological reporting. In patients without surgery (non-appendicitis) the reference standard was follow-up after initial assessment to rule out appendicitis with an ideal standard of ≥14 days with active follow-up. Studies that differed or were unclear about confirming these standards would be highlighted when assessing risk of bias.

Studies were included if they matched these predefined criteria and reported sufficient data to construct or derive a two-by-two contingency table for any/multiple blood tests.

Studies were excluded if they only reported surgical cases, had <20 participants, reported atypical appendiceal pathology (eg, carcinoid, appendiceal mass) or if the proportion of participants outside the age range was >15%. Where multiple studies reported data on the same population, only outcomes of the earliest study were used, unless the authors confirmed individual patient results.

Screening of titles and abstracts was performed independently by two reviewers (DF-C and GAH). Full-text articles of all short-listed studies were then assessed for eligibility.

Four reviewers (DF-C, GAH, JMO-M and MA) independently extracted data from each included study on a predesigned data extraction sheet including information on demographics, prevalence, reference standards, measurement of blood tests including cut-offs and 2×2 tables. Disagreements were resolved by consensus or from input of an independent reviewer (GH).

In studies which matched inclusion criteria exclusion with reason would occur if: (A) data were not able to be extracted in 2×2 format (derivation from reported sensitivity/specificity or other metrics would be used if

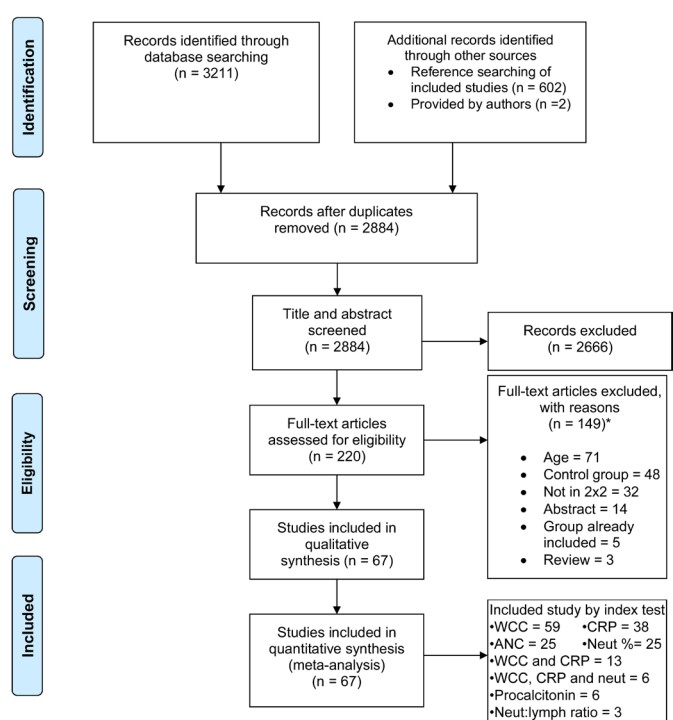

**Figure 1** PRISMA flow chart for identification of included studies. *Some studies excluded for two reasons, see full overview of excluded studies in online supplemental table 1.[24] ANC, absolute neutrophil count; CRP, C reactive protein; PRISMA, Preferred Reporting Items for Systematic Reviews and Meta-Analyses; WCC, white cell count.

possible); (B) a paediatric subgroup was not adequately detailed; (C) exclusion criteria were unclear and (D) the paper was only available in an abstract form. If reason (A) was given for full-paper exclusion an attempt would be made to contact an author on up to three occasions before exclusion. If contacted authors provided individual patient data (IPD), this was included.

### Assessment of study methodological quality

All included studies were assessed by four reviewers (DF-C, GAH, JMO-M and MA) independently for risk of bias and applicability using the QUADAS-2 tool.[25] Discrepancies were resolved by discussion or review from a third independent reviewer (GH) to achieve consensus. Full details of QUADAS-2 scoring criteria are detailed in online supplemental material 1: QUADAS-2. Overview graph of QUADAS-2 outcomes was created using GraphPad Prism (V.8.0, GraphPad Software).

### Data synthesis and statistical analysis

We extracted binary data for all appropriate measurements of diagnostic accuracy for the predefined blood tests. Cut-offs were defined by authors in each individual publication. In cases where individual data was provided by authors a 2×2 table was calculated using the most commonly reported cut-offs. Data for age-dependent cut-offs would be included in study overviews and forest plots but not in cut-off specific meta-analysis. Measurement

of neutrophils by means of absolute count (ANC) or percentage of white cells was analysed separately.

Test accuracy overviews for each marker were reported as forest plots including 2×2 tables, sensitivity and specificity with 95% confidence intervals using RevMan (Cochrane collaboration, Copenhagen, Denmark). Meta-analysis was performed using a bivariate random effects model and pooled estimates of sensitivity, specificity and likelihood ratios were generated when >4 studies per marker was included using the metandi module in STATA V.16.[26–28] When a study provided multiple cut-offs for each test, we calculated the summary receiver operating characteristics (SROC) curve with 95% CI using R diagmeta package developed by Steinhauser *et al*[29] assuming a fixed intercept (diagnostic accuracy) and slope (threshold effect) across studies. In addition, we calculated the pooled sensitivity and specificity for commonly presented cut-offs and we compared the results with the bivariate random effects model of cut-off groups of these tests.

We assessed the heterogeneity by visual inspection of the forest plot and SROC plot. Potential differences in studies with: high risk of bias in patient selection; case–control design; from primary care; with low prevalence (<10%); with high prevalence (≥60%), were evaluated by adding these covariates to the bivariate model and performing sensitivity and subgroup analysis to measure if these factors could explain heterogeneity. To assess for publication bias we used the Deeks' test.[30]

### Potential clinical impact

To provide more insight into the clinical consequences of using the results of each test, for each measurement with pooled sensitivity and specificity hypothetical 2×2 tables were constructed in 1000 children presenting with symptoms suggestive of acute appendicitis. The appendicitis prevalence was modelled on the median prevalence in the cohort studies included in the final meta-analysis, rounded to the nearest 10% to help facilitate interpretation. Standardisation of prevalence allowed comparison of the results of each test.

To illustrate the clinical application of tests we interpreted these as follows: children with appendicitis missed are those with appendicitis and a negative index test result (false negatives); the numbers of unnecessary surgeries are the children without appendicitis with a positive index test result (false positives).

### RESULTS
### Search strategy and study selection

The search strategy, reference screening and paediatric data provided by contacted authors identified 3211 studies. Sixty-seven studies matched inclusion criteria and were included in final analysis, of which 11 studies had patient data provided after contacting authors to clarify criteria or provide paediatric-only data.[31–41] Remaining papers were excluded with reason (figure 1, online supplemental table 1).

## Study characteristics

All included studies were in the hospital or ED setting. The most common study designs were cohort study, with either a prospective (n=40, 60%) or retrospective (n=11, 16%) cohort design or case control (n=6, 9%) (table 1, online supplemental table 2). One study had a low prevalence (<10%) of appendicitis.[42] Seven cohort studies had a high prevalence (>60%) due to a more select setting such as GP referrals,[43 44] hospitalised[39 43 45 46] or ITU based patients.[41] In one study the reason for high prevalence was unclear.[47] The median prevalence for all studies was 44% (IQR 34%–58%). No study used POC blood test devices.

The most frequent reference standard in appendicitis patients was histopathology with the remainder using operative reports (table 1, online supplemental table 3). Of 67 studies, 61 (91%) described inclusion of a population of children with clinical symptoms of appendicitis and eight of these studies specified a blood test and/or ultrasound for inclusion; the remaining 6 studies (9%) were of case–control designs and most commonly identified cases through surgery and controls as those with symptoms of appendicitis which had the condition ruled out (online supplemental table 2).

## Risk of bias

Applying the QUADAS-2 framework,[25] patient selections in 60 studies (90%) were low risk of concern for applicability of results as they clearly included patients with suspected acute appendicitis. Those not scoring low included higher ages (n=4)[36 48–50] or only those undergoing ultrasound scans (8.3%, n=5).[51–55] However fewer studies scored low for risk of bias (44%, n=30) when considering patient selection; most commonly because studies did not clearly state whether they recruited a random or fully consecutive population (figure 2, online supplemental table 4).

Risk of bias and applicability of the index test and reference standard was often low risk or unclear—owing to a lack of detail regarding blinding (ie, clinician or pathologist) or follow-up method. No study scored low for 'flow and timing' as a component as this required both groups to receive the same reference standard. It is understandably not ethical to perform surgery in all children, and therefore, in this dimension, the seven studies that score 'unclear' due to lack of detailed outcomes for all participants are likely of greater concern (figure 2).

## Blood tests

Sixty-seven included studies consisted of 34 839 children (13 342 with appendicitis and 21 497 without) which evaluated the following tests: WCC, CRP, ANC, neutrophils as percentage of leucocytes, procalcitonin, combined WCC and CRP, and combined WCC/CRP/neutrophil percentage. Neutrophil to lymphocyte ratio was reported in three studies which was not sufficient to allow a meta-analysis. Reported accuracies ranged from sensitivity 0.18–0.91, and specificity 0.69–0.95 but cut-offs used

**Table 1** Pooled overview of design and standards of included studies

|  | Studies (n=67) |
| --- | --- |
| **Study design** | |
| Prospective cohort study | 40 (60%) |
| Retrospective cohort study | 11 (16%) |
| Case control study | 6 (9%) |
| Prospective observational study | 3 (4%) |
| Retrospective case series | 2 (3%) |
| Other* | 5 (7%) |
| **Setting** | |
| Institution | |
| Hospital | 67 (100%) |
| Department† | |
| Emergency department | 49 (73%) |
| Surgical department | 17 (25%) |
| Centre | |
| Single centre | 54 (81%) |
| Multiple centre | 13 (19%) |
| **Study participants** | |
| Total (n) | 34 839‡ |
| Median patients included (range) | 275 (44–3791) |
| ≤50 participants | 4 (6%) |
| 51–250 participants | 28 (42%) |
| 251–500 participants | 14 (21%) |
| >500 participants | 21 (31%) |
| **Year published** | |
| Before 1989 | 2 (3%) |
| 1990–1999 | 5 (7%) |
| 2000–2009 | 14 (21%) |
| 2010–2022 | 46 (69%) |
| **AA prevalence** | |
| <20% | 6 (9%) |
| 21%–35% | 14 (21%) |
| 36%–50% | 24 (36%) |
| 51%–65% | 18 (27%) |
| >65% | 5 (7%) |
| **Gender demographics (% male)** | |
| <35% | 1 (1%) |
| 36%–45% | 8 (12%) |
| 46%–55% | 29 (43%) |
| 56%–65% | 20 (30%) |
| >65% | 2 (3%) |
| Unclear | 7 (10%) |
| **Reference standard (AA)** | |
| Histology/pathology | 63 (94%) |
| Operative findings | 4 (6%) |

Continued

**Table 1** Continued

| | Studies (n=67) |
|---|---|
| Reference standard (non-AA) | |
| Active follow-up | 19 (28%) |
| Discharge diagnosis | 16 (24%) |
| Unspecified follow-up | 8 (12%) |
| Observation | 8 (12%) |
| Unclear | 8 (12%) |
| Failure to reattend | 6 (9%) |
| Case note review | 2 (3%) |
| Index test(s) measured | |
| White cell count (WCC) | 59 (88%) |
| C reactive protein (CRP) | 38 (57%) |
| Absolute neutrophil count | 25 (37%) |
| Neutrophil percentage | 16 (24%) |
| WCC and CRP | 13 (19%) |
| WCC, CRP and neutrophil | 6 (9%) |
| Procalcitonin | 6 (9%) |
| Neutrophil:lymphocyte ratio | 3 (4%) |
| WCC and Neutrophil count | 1 (1%) |

Full details of all study characteristics including design, setting, population characteristics and demographics, exclusion and inclusion criteria is fully detailed in online supplemental table 2. Full details of study reference standard, index tests and cut offs in online supplemental table 3.
*In 5 studies authors reported other study designs than those listed.
†In one study (1%) department setting was unclear.
‡Value includes all participants in included studies, in five studies a paediatric subgroup was provided meaning total paediatric patients in analysis n=31 143.

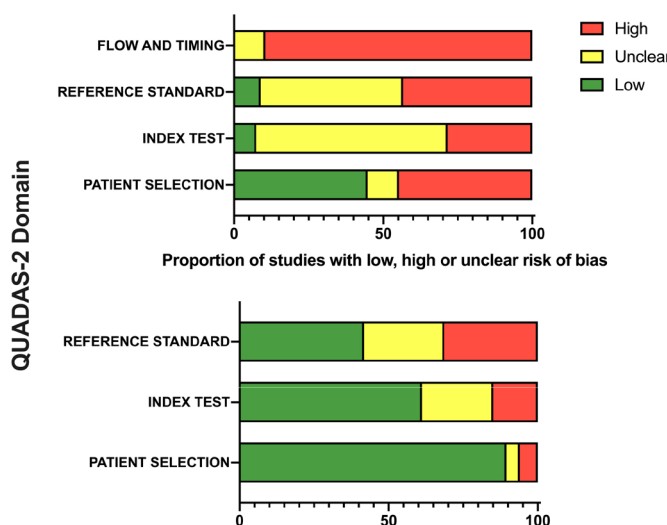

**Figure 2** Overview of QUADAS-2 assessment of all included studies (n=67) showing review authors' assessment of risk of bias and study applicability in each domain reported as a percentage of all included studies (proportion coloured by assessment as high (red), unclear (yellow) or low (green), full outline of all individual components of QUADAS-2 is detailed in online supplemental table 4.

were dissimilar.[56–58] Similarly, the combination of WCC and ANC (>13 300/µL and >10 800/µL respectively) was reported in one study (sensitivity=0.76, specificity=0.82).[59] We found no evidence of publication bias for each test where this was appropriate to assess (online supplemental figure 1).

An overall pooled estimate for each test, at different cut-offs, is shown in table 2. For WCC and CRP, the pooled estimates obtained for each cut-off with the Steinhauser method were similar for the cut-off groups using the bivariate model (online supplemental table 5). In subgroup analyses, the test characteristics did not differ between studies with a high appendicitis prevalence (≥60%) and lower prevalence. In some tests when using bivariate analysis the study design (for WCC 10–15 000 cells/µL and neutrophils) or patient selection (for WCC and CRP) had an effect on the sensitivity or specificity which we quantified (online supplemental table 5).

### White cell count
WCC was the most frequently reported blood test (59 studies, including 22 779 patients; 9733 with appendicitis).

At a cut-off of 10 000 cell/µL WCC exhibited one of the highest pooled sensitivities of any individual test (0.85, 95% CI 0.80 to 89, 53 studies) but specificity was low (0.58, 95% CI 0.51 to 0.64) table 2, figure 3A, (online supplemental figure 2).

### C reactive protein
CRP was the second most frequently reported test (38 studies, 9934 patients (4201 appendicitis). The sensitivity and specificity for lower cut-off values varied considerably between studies on visual inspection (online supplemental figure 3). Specificity of a CRP value of ≥50 mg/L was more homogeneous. The highest pooled sensitivity of CRP was at a cut-off of ≤5 mg/L (0.77 (95% CI 0.69 to 0.84)), although specificity was low (0.54, 95% CI 0.45 to 0.63, table 2). Use of a cut-off ≥50 mg/L had the best specificity of any single test at 0.87 (95% CI 0.80 to 0.91, table 2 and figure 3B) and all studies reported a specificity of >0.8 for this (online supplemental figure 3).

### Neutrophil measurement
Neutrophil measurement was reported as an absolute count (ANC) in 25 studies, including 12 982 patients (5311 appendicitis). ANC was more frequently reported and demonstrated the highest sensitivity of any test at a low cut-off (≥7500 cells/µL, 0.90, 95% CI 0.85 to 0.94) table 2, (online supplemental figure 4). However, none of the studies evaluating ANC reported multiple cut-offs and therefore the SROC curve could not be calculated using the Steinhauser method. Relatively more studies (n=13) reported a cut-off of 7500 cells/µL which reported a similar sensitivity (0.86 (95% CI 0.82 to 0.89)), table 2 to WCC.

**Table 2** Pooled estimates of diagnostic accuracy of makers for diagnosis of appendicitis in children

| Marker | Studies | Threshold | Pooled sensitivity* (95% CI) | Pooled specificity* (95% CI) | Hypothetical cohort of 1000 children with appendicitis prevalence of 40%† | | Hypothetical cohort of 1000 children with appendicitis prevalence of 5% | |
| --- | --- | --- | --- | --- | --- | --- | --- | --- |
| | | | | | Missed appendicitis (FN) | Unnecessary treatment (FP) | Missed appendicitis (FN) | Unnecessary treatment (FP) |
| WCC‡ (cells/µL) | 53 | 10000 | 0.85 (0.80 to 0.89) | 0.58 (0.51 to 0.64) | 60 | 252 | 8 | 399 |
| | | 12000 | 0.73 (0.66 to 0.79) | 0.71 (0.65 to 0.76) | 108 | 174 | 14 | 276 |
| | | 15000 | 0.53 (0.45 to 0.62) | 0.83 (0.78 to 0.87) | 188 | 102 | 24 | 162 |
| CRP§ (mg/L) | 38 | 5 | 0.77 (0.69 to 0.84) | 0.54 (0.45 to 0.63) | 92 | 276 | 12 | 437 |
| | | 10 | 0.68 (0.59 to 0.76) | 0.66 (0.58 to 0.73) | 128 | 204 | 16 | 323 |
| | | 30 | 0.50 (0.39 to 0.60) | 0.81 (0.75 to 0.87) | 200 | 114 | 25 | 181 |
| | | 50 | 0.41 (0.30 to 0.53) | 0.87 (0.80 to 0.91) | 276 | 78 | 35 | 124 |
| Neutrophils¶ (% of lymphocytes) | 12 | 75% | 0.76 (0.73 to 0.80) | 0.63 (0.57 to 0.69) | 96 | 222 | 12 | 352 |
| ANC (cells/µL) | 5 | <7500 | 0.90 (0.85 to 0.94) | 0.54 (0.49 to 0.58) | 40 | 276 | 5 | 437 |
| | 13 | 7500 | 0.86 (0.82 to 0.89) | 0.60 (0.54 to 0.66) | 56 | 240 | 7 | 380 |
| | 6 | >7500 | 0.70 (0.65 to 0.75) | 0.78 (0.71 to 0.85) | 120 | 132 | 15 | 209 |
| Procalcitonin (ng/mL) | 6 | 0.1–0.5 | 0.30 (0.26 to 0.35) | 0.83 (0.73 to 0.89) | 280 | 102 | 35 | 161 |
| WCC and CRP** | 6 IPD studies | 10000 and 10 | 0.59 (0.49 to 0.68) | 0.79 (0.74 to 0.84) | 164 | 126 | 21 | 200 |
| WCC and CRP | 6 IPD studies | 10000 and 50 | 0.27 (0.19 to 0.37) | 0.93 (0.91 to 0.95) | 292 | 46 | 37 | 67 |
| WCC or CRP | 6 IPD studies | 10000 and 10 | 0.97 (0.93 to 0.99) | 0.40 (0.30 to 0.51) | 12 | 360 | 2 | 570 |
| WCC or CRP | 6 IPD studies | 10000 and 50 | 0.91 (0.86 to 0.95) | 0.50 (0.40 to 0.60) | 36 | 300 | 5 | 475 |

Six studies could not be used in the pooled analysis, because 5 studies reported WCC provided age-dependent cut offs and 1 study reported pain duration dependent cut offs.

*All pooled sensitivity and specificity were obtained from the bivariate model, except for WCC and CRP for which we calculated sensitivity and specificity for selected cut offs using the Steinhauser method, comparison with bivariate mode including likelihood ratios for all tests is demonstrated in online supplemental table 5.

†Median prevalence of appendicitis in the included 62 cohort studies is 40%.

‡WCC was evaluated in 59 studies whereof 53 were included in the meta-analysis whereof 12 provided multiple cut offs.

§CRP was evaluated in 39 studies whereof 11 provided multiple cut offs.

¶Neutrophils evaluated in 16 studies of which 12 provided cut off at 75%.

**WCC and CRP calculated from IPD from 6 studies (n=1541 participants) and analysed using the bivariate model).

***WCC and CRP calculated from IPD from 6 studies (n=1541 participants) and analysed using the bivariate model).

ANC, absolute neutrophil count; CRP, C reactive protein; FN, False Negatives; FP, False Positives; IPD, individual patient data; WCC, white cell count.

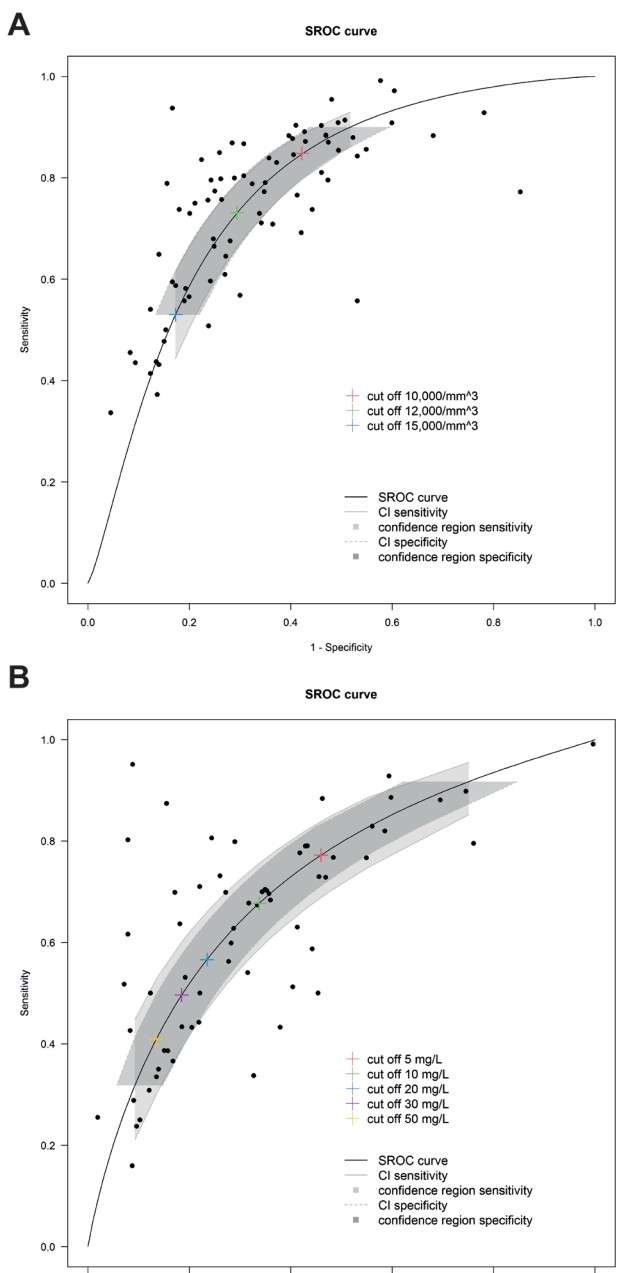

**Figure 3** Summary receiver operating characteristic plot of WCC (A) and CRP (B) for diagnosis of paediatric acute appendicitis (colour of dot representing study identified in legend, hashed grey box=95% CI specificity, solid grey box=95% CI sensitivity. Examples of commonly reported cut-offs shown in (A) as blue (10 000/μL), green (12 000/μL) and red (15 000/μL) crosses on curve; in (B) as yellow (5 mL/L), pink (10 mg/L), blue (20 mg/L), green (30 mg/L) and red (50 mg/L) crosses on curve. CRP, C reactive protein; SROC, summary receiver operating characteristics; WCC, white cell count.

Fewer studies (n=16, 5884 patients; 2275 appendicitis) reported neutrophil count as a percentage (online supplemental figure 5), (online supplemental table 5), the most common cut-off reported was 75% and pooled

(n=9) analysis at this level showed the sensitivity to be lower than ANC and WCC (table 2).

### Procalcitonin

Procalcitonin accuracy was quantified in 6 studies (860 patients; 469 appendicitis), which led to pooled analysis being performed on varied cut-offs. Sensitivity was homogeneous, while specificity varied between the studies (online supplemental figure 6). Pooled procalcitonin had a relatively high pooled specificity (0.83, 95% CI 0.73 to 0.89) but a low sensitivity (0.30, 95% CI 0.26 to 0.35) (table 2).

### Combinations of tests

The cut-offs of tests in combination did vary between studies and the studies were therefore too heterogeneous to perform a meta-analysis. A combination of CRP and WCC (11 studies, 2497 patients, 1008 appendicitis) demonstrated a sensitivity range of 0.34–0.80 and specificity range of 0.50–0.95 (online supplemental figure 7). A combination of one test being positive (10 studies, 5490 patients, 1912 appendicitis) showed a sensitivity and specificity range of 0.85–1.00 and 0.07–58, respectively.

A combination of CRP, WCC and neutrophils (five studies, 1410 patients, 506 appendicitis) showed a range of sensitivities (0.43–0.86) but specificity was generally high (0.81–0.91) and one study evaluated the *or* combination with a sensitivity of 1.00 (95% CI 0.97 to 1.00) and specificity of 0.25 (95% CI 0.20 to 0.30) (online supplemental figure 8).

To account for pooling of differing cut-offs in combinations of tests we performed a subgroup analysis of patients from six studies which provided IPD on request for clarification.[31–35 41] This included the raw results for 1541 patients and showed that a combination of WCC >10 000 cell/μL and CRP >10 mg/L has reasonable pooled specificity (0.79, 95% CI 0.74 to 0.84), which was increased when the threshold for CRP was raised to >50 mg/L with WCC (0.93, 95% CI 0.91 to 0.95), with a resulting low sensitivity (0.27, 95% CI 0.19 to 0.37, table 2). Using a combination of tests as either WCC or CRP being positive at a low cut-off generated the best sensitivity of any measured investigation (WCC>10 000 cell/μL or CRP>10 mg/L; 0.97, 95% CI 0.93 to 0.99, table 2).

### Clinical impact

Modelling the effect of each test on a hypothetical cohort of 1000 children with a 40% prevalence of appendicitis, which was the median prevalence of all included studies rounded to the nearest decile (median 44%, IQR 33%–58%), showed that the best single tests performing best for ruling out appendicitis were ANC <7500 cells/μL and WCC ≥10 000 cells/μL, as these only miss 40 and 60 of the 400 children with appendicitis, respectively, but would capture a large number of non-appendicitis cases for further review of treatment—276 and 252 out of 600, respectively (table 2, online supplemental table 5). Therefore, in clinical application these single tests would

be the best for supporting a decision to discharge, or to not refer a patient for surgical review.

The best single test for ruling in appendicitis was a CRP value ≥50 mg/L and so could be the most useful in deciding to rule in the diagnosis of appendicitis. Although in the hypothetical cohort 78 would still undergo unnecessary review or treatment (False positive) of the 202 that would meet this threshold. Also, 276 children with appendicitis would be missed (table 2).

Using IPD identified a combination of WCC or CRP (WCC >10 000 cell/µL or CRP >10 mg/L) to lead to the least cases of missed appendicitis (n=12). Similarly, a combination of WCC >10 000 cells/µL and CRP >50 mg/L was more specific, application of this to inform surgical decision would result in the lowest number of unnecessary treatment (n=46) but only 150 patients would meet this threshold in the hypothetical cohort, 104 of which would have appendicitis (table 2).

Although no included studies were in the community setting, the analysis was repeated with a lower prevalence (5%), which is reported as the prevalence in primary care.[60] This was used to reflect metrics for use of blood tests in a non-ED setting, or if it was used for a triage test to decide other diagnostic modalities such as ultrasound (table 2).

## DISCUSSION

In a pooled analysis of 31 143 children attending hospital with symptoms of appendicitis the most sensitive single blood tests for discriminating appendicitis from non-surgical conditions were WCC or ANC, and combining WCC with CRP increases this further. The most specific were CRP with or without WCC. However, if these were used as a sole decision-making tool for discharge in a hypothetical cohort of 1000 children with 40% appendicitis prevalence, even the most sensitive single test would result in 40 cases of missed appendicitis (ANC).

Conversely, although a blood test alone would unlikely be used as the sole tool for surgical decision making, if a blood test was used to rule in the condition the most specific test would result in 78 unnecessary treatments (CRP). Therefore, all common inflammatory blood tests have limitations in the diagnosis of paediatric appendicitis which we accurately quantify. We highlight which tests should be valued most highly in more complex diagnostic pathways.

### Strengths

To our knowledge, this is the largest systematic review of the diagnostic accuracy of commonly utilised blood tests to support diagnosis in children with suspected appendicitis. Through use of a broad search strategy we combined information from 67 studies and report pooled measurements for five common blood tests at multiple cut-offs and in combination. Our analysis used multiple models for quantification and also undertook analysis for potential subgroup and publication biases. Previous

diagnostic accuracy meta-analysis methods have used diagnostic data for a single cut-off, and often the selected cut-off is the one maximising both diagnostic accuracy parameters (sensitivity and specificity). In our methods we used commonly reported cut offs to identify those with the highest clinical impact. Thus, by using our approach, we ensure the pooled diagnostic accuracy parameters are not overestimated.[29]

The study gained important additional data through contacting authors and used IPD where possible, adding to the applicability of results to clinical practice. It also aided us in reporting combination of tests, with cut-offs being variable in literature but allowing inclusion of over 1500 children to demonstrate the power of combined WCC and CRP to maximise sensitivity if either are positive, or specificity if both are negative. The included studies had representation from a variety of continents and 69% were undertaken in the last twelve years adding to the applicability of results. In addition, 94% of the studies confirmed the diagnosis of acute appendicitis with histopathology results adding to the reliability of findings.

### Potential limitations

An important limitation is that no studies were identified from the primary care settings, or reported results of POC devices. Our findings could be readily applicable to the emergency setting for use in triage, similar to other conditions.[61] Although our sensitivity analysis suggested that prevalence did not affect the diagnostic accuracy, these results should be interpreted with care as there was only one study with a very low prevalence (<10%).[42] We repeated the cohort analysis in a lower prevalence setting (5%) to illustrate likely performance if blood tests were used in the community or as a triage test for imaging investigation, although metrics did reflect information from higher prevalence studies and differences in casemix or severity in appendicitis could result in a different diagnostic accuracy.

Another possible limitation is that acute appendicitis can represent a spectrum of severity, some reports suggesting histological subtypes of appendicitis can result in differing laboratory results.[62] This differentiation can only be made postoperatively, but our approach was to evaluate test application to all cases of suspected appendicitis so that it best represented the diagnostic decision-making when clinicians first encounter cases. A small number of studies (n=5 for WCC, n=1 for ANC and n=1 for neutrophils) also divided tests thresholds by age. These were too few to pool and were heterogeneous, so these results were reported but not used in meta-analysis.

In addition, pooled metrics were applied to a hypothetical cohort which aids in the application but would not wholly represent the clinical setting, where full decision making would be more complex and could use other diagnostic modalities such as imaging. Also, although these results give one of the largest pooled analysis of blood tests the strategy is not able to identify the added value

of these to specific signs and symptoms, or as a triage to additional diagnostics.

## Comparison with literature

Smaller systematic reviews including adults and children have reported different metrics for diagnostic accuracy, such as WCC sensitivity being 0.62–0.87,[18 63 64] which is likely a result of pooling multiple cut-offs from studies. To mitigate this effect we have pooled by similar cut-off generating SROC curves for each or from IPD where possible. This should increase the reliability and applicability of results, and has highlighted that at a specific cut-off of WCC sensitivity was higher than previously reported (0.84 at 10 000 cells/μL).

Other studies report combined clinical scoring tools for diagnosing appendicitis, with blood tests as one component. A recent review reported the best performing of these (the Shera score) at a low cut-off to miss 18 out of 539 cases of paediatric appendicitis if used.[20] The specificity of this scoring tool was lower than any of our measured blood tests however (44.3%, 95% CI 41.4% to 47.2%). These tools have the advantage of incorporating clinical signs and symptoms, and our results complement these by quantifying the exact benefit of tests at different cut-offs.

Blood tests in paediatric appendicitis have also been reviewed as part of a systematic review of other diagnostic modalities, with WCC identified as sensitive at 10 000 cells/μl (0.88%, 95% CI 87% to 90%) and CRP as specific (no pooled analysis).[65] This did not include retrospective or non-ED studies, being limited to single studies at some cut offs. Our study seems to agree with this but included more studies and IPD where available which allowed reporting of different cut-offs and highlighted the benefit of test combinations.

## Clinical implications

Applying the meta-analysis results to a hypothetical clinical cohort demonstrated that ANC, WCC and CRP are the tests with greatest clinical value. Although in practice clinicians may not rely on blood tests alone, it is still important to know ANC or WCC (alone or combined with CRP) at a low cut-off would be the best tests to inform a decision to discharge. Using this strategy for triage would result in a relatively low number of missed appendicitis cases (60 and 40 out of the 400 cases in the modelled cohorts respectively). Combining WCC and CRP increases this further (12/400 missed). These values could be used to develop triage pathways, or equate risk to parents to self-observe. These results could provide the basis for application of selected blood tests in a prospective trial in the triage setting, or an evaluation of POC tests outside the ED.[66]

CRP at a high cut-off or combined with WCC was the most specific test but a significant proportion of children would still reach this threshold, and a large number of true cases would be missed (276 and 188, respectively). A recent international cohort study of over 1800 children with right iliac fossa pain, identified a negative appendicectomy rate of 15.9%, increasing to 22.4% in girls aged 11–15 years.[20] Our results suggest using those blood tests to inform surgery would perform worse than this, highlighting blood tests alone should not be used to rule in appendicitis.

Application of these results, especially in the situation of combined blood tests in addition to signs and symptoms, would leave a proportion of children between the two clinical situations of a positive low WCC/ANC and high CRP. In these children where diagnostic uncertainty remains, further assessment by specialist or use of an adjunct such as ultrasound could be considered. Using specific tests in combination with prediction rules to determine the need for investigations such as ultrasound could also lead to an increased diagnostic accuracy of imaging, being higher when the pretest probability increases.[67]

Our results would not support using procalcitonin or neutrophil percentage as a preferred test for clinical decision-making.[68]

## CONCLUSION

In conclusion, we report the largest study of commonly available blood tests and identify WCC value below 10 000 cells/μL and ANC value below 7500 cells/μL are the best single tests in ruling-out suspected childhood appendicitis. Combining WCC and CRP (10 000 cells/μL or 10 mg/L) increased this further. These tests could be used in addition to signs and symptoms to discharge or safety-net patients. A CRP value above 50 mg/L in addition to signs and symptoms could be used to select patients for further imaging. However, more information about the exact added value of these blood tests incorporated with other diagnostic modalities is needed. Since we did not find any primary care studies or studies evaluating POC tests, these three tests could be prioritised for future evaluation.

**Author affiliations**

[1]NIHR Community Healthcare MedTech and IVD Co-operative, Nuffield Department of Primary Care Health Sciences, University of Oxford, Oxford OX2 6GG, UK
[2]Academic Paediatric Surgery Unit, Nuffield Department of Surgical Sciences, University of Oxford, Oxford OX3 9DU, UK
[3]MRC Weatherall Institute of Molecular Medicine, Radcliffe Department of Medicine, University of Oxford, Oxford OX3 9DS, UK
[4]Department of General Practice and Elderly Care Medicine, University of Groningen, University Medical Center Groningen, Groningen, The Netherlands
[5]EPI-Centre, Academic Centre for Primary Care, Department of Public Health and Primary Care, KU Leuven, 3000 Leuven, Belgium
[6]NIHR Oxford Biomedical Research Centre, Oxford University Hospitals NHS Foundation Trust, Oxford OX3 9DU, UK

**Acknowledgements** We thank Nia Roberts and the Bodleian healthcare libraries for assistance in conducting the search strategy. We acknowledge help in the review of foreign language studies provided by Patrick Garfjeld Roberts, University of Oxford. We appreciate those authors that provided patient data on request to allow inclusion of their original studies

**Contributors** DF-C, GH, AVDB and GAH conceived and designed the study. DF-C and GAH assessed initial search results and selected studies for inclusion with discrepancies resolved with GH and AVDB. DF-C, GH, JMO-M and MA performed data extraction. DF-C and GAH performed analysis. All authors were involved in formation and revision of final manuscript. The corresponding author attests that all listed authors meet authorship criteria and that no others meeting the criteria have been omitted. DF-C is the guarantor of the study.

**Funding** This research was funded by the National Institute for Health and Care Research (NIHR) Community Healthcare MedTech and In Vitro Diagnostics Co-operative at Oxford Health NHS Foundation Trust. The views expressed are those of the author(s) and not necessarily those of the NHS, the NIHR or the Department of Health and Social Care. DF-C received funding from the National Institute for Health Research as an Academic Clinical Fellow and the Wellcome Trust Doctoral Training Fellowship at the University of Oxford. GAH received funding by Ter Meulen Grant of the Royal Netherlands Academy of Arts and Sciences, and NWO talent programme Veni.

**Competing interests** None declared.

**Patient and public involvement** Patients and/or the public were involved in the design, or conduct, or reporting, or dissemination plans of this research. Refer to the Methods section for further details.

**Patient consent for publication** Not applicable.

**Provenance and peer review** Not commissioned; externally peer reviewed.

**Data availability statement** Data sharing not applicable as no datasets generated and/or analysed for this study.

**ORCID iDs**
David Fawkner-Corbett http://orcid.org/0000-0002-5273-3682
Gail Hayward http://orcid.org/0000-0003-0852-627X
Mohammed Alkhmees http://orcid.org/0000-0002-8608-6938
Ann Van Den Bruel http://orcid.org/000-0001-9012-2009
Jose M Ordóñez-Mena http://orcid.org/0000-0002-8965-104X
Gea A Holtman http://orcid.org/0000-0001-6579-767X

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
