## [Reviewer comments · BMJ Open]

ARTICLE DETAILS

TITLE (PROVISIONAL)	Diagnostic accuracy of blood tests of inflammation in paediatric appendicitis – a systematic review and meta-analysis
AUTHORS	Fawkner-Corbett, David; Hayward, Gail; Alkhmees, Mohammed; Van Den Bruel, Ann; Ordóñez-Mena, JM; Holtman, G.

VERSION 1 – REVIEW

REVIEWER	Yong, Chen KK women's and children's hospital , pediatric surgery
REVIEW RETURNED	08-Nov-2021

GENERAL COMMENTS	The paper reports a systematic review and meta-analysis of diagnostic accuracy of serum inflammation markers for paediatric appendicitis based on 60 studies (28,0001 children). Author concluded that the best performing blood tests for ruling-out pediatric appendicitis are WCC or ANC (sensitivity 0.87 and 0.90 respectively). The most specific tests were CRP alone ($\geq 50\text{mg/l}$, 34 studies, specificity 0.87[95% CI 0.81-0.91]) or CRP combined with WCC ($\geq 10,000$ cells/μl, six studies 0.93[95% CI 0.91-0.95]. Questions/comments for the authors: 1. To describe the performance of a diagnostic test, the positive and negative predictive values (PPV and NPV) are more relevant than sensitivity and specificity. Authors should added PPV (the chance of appendicitis if the test is positive) and NPV (the chance of non-appendicitis if the test is negative) in the table/figures.2. WCC, ANC, CRP and procal are actually non-specific inflammation markers. A positive blood test suggests the presence of inflammation but it would be difficult to differentiate appendicitis from other inflammatory conditions (such as bacteria gastroenteritis). Therefore, the above tests have a limited role in diagnosing appendicitis but can be used for ruling-out appendicitis.3. As per what the authors described , the best performing blood tests for ruling-out pediatric appendicitis are WCC or ANC with sensitivities 0.87 and 0.90 respectively. However, more than 10 % of appendicitis would have been missed if WCC or ANC alone are used for diagnosis. A few papers have demonstrated that the combination of inflammatory markers has a dramatically improved diagnostic value for appendicitis [1–3]. The combined sensitivity of WCC, ANC and CRP is close to 100% [4]. In this study, the pooled sensitivity for combined WCC, ANC and CRP is only 80%, which could be due to the difference in interpreting combined test results.4. The methods section did not specify what constitutes a positive combined test. Does a positive combined test require all 3
---

	inflammation markers to exceed their cut-off values or any of the 3 inflammation markers exceed their cut-off value? The sensitivity and NPV would only be raised if the positive combined test is defined as ANY of the inflammation markers exceed their cut-off value. Chiang, et al found that WCC and ANC are more sensitive in early appendicitis , while the sensitivity of CRP is higher in late/complicated appendicitis. 99.7% of appendicitis has at least one of the inflammation markers raised[4]. In other words, if patients have normal WCC,ANC and CRP, the risk of appendicitis is extremely low (<0.3%). The author should relook into the studies for combined test and check if a negative combined test (all makers are normal) is a strong indicator for ruling-out appendicitis. 5. The searching is not up to date (4th June 2019). Studies from the last 2 years should be included in this meta-analysis. 6. Small error: sup table 5, last two rows are duplicated [1] Birchley D. Patients with clinical acute appendicitis should have pre-operative full blood count and C-reactive protein assays. Ann R Coll Surg Engl 2006;88:27–32. doi:10.1308/003588406X83041. [2] Grönroos JM, Grönroos P. Leucocyte count and C-reactive protein in the diagnosis of acute appendicitis. Br J Surg 1999;86:501–4. doi:10.1046/j.1365-2168.1999.01063.x. [3] Yap T-L, Chen Y, Low WWX, Ong CCP, Nah SA, Jacobsen AS, et al. A new 2-step risk-stratification clinical score for suspected appendicitis in children. J Pediatr Surg 2015;50. doi:10.1016/j.jpedsurg.2015.08.028. [4] Chiang JJY, Angus MI, Nah SA, Jacobsen AS, Low Y, Choo CSC, et al. Time course response of inflammatory markers in pediatric appendicitis. Pediatr Surg Int 2020;36:493–500. doi:10.1007/S00383-020-04620-4.
--	---

REVIEWER	Hajibandeh, S Royal Glamorgan Hospital
REVIEW RETURNED	11-Nov-2021

GENERAL COMMENTS	Many thanks for the opportunity to review the manuscript titled “Diagnostic accuracy of blood tests of inflammation in paediatric appendicitis – a systematic review and meta-analysis”. The authors conducted a systematic review and meta-analysis of 60 studies enrolling a total of 28,001 paediatric patients and concluded that white cell count (WCC) and absolute neutrophil count (ANC) are the best performing blood tests for ruling-out paediatric appendicitis. Overall, the study is very well-designed and well-conducted with satisfactory methodological and reporting qualities. The comparison of diagnostic accuracy of WCC, ANC, CRP, and procalcitonin in children with appendicitis is not novel anymore and has been extensively studied in the past as reflected by the number of studies included in this study; indeed they are commonly used in practice. Nevertheless, the main question is whether the other biomarkers which are not commonly used or have never been used for diagnosing acute appendicitis are better predictors than the commonly used ones. The recent evidence suggests that Neutrophil-to-Lymphocyte Ratio (NLR) is a very promising biomarker that can predict both diagnosis and severity of appendicitis. It is very surprising that the applied search strategy by the authors could only identify one study on diagnostic accuracy of NLR in children with acute appendicitis. A very quick
---

search identified the following articles on diagnostic accuracy of NLR in children with acute appendicitis:

- Prasetya D, Rochadi, Gunadi. Accuracy of neutrophil lymphocyte ratio for diagnosis of acute appendicitis in children: A diagnostic study. *Ann Med Surg (Lond)*. 2019 Oct 17;48:35-38.
- Sengul S, Guler Y, Calis H, Karabulut Z. The Role of Serum Laboratory Biomarkers for Complicated and Uncomplicated Appendicitis in Adolescents. *J Coll Physicians Surg Pak*. 2020 Apr;30(4):420-424.
- Tuncer AA, Cavus S, Balcioglu A, Silay S, Demiralp I, Calkan E, Altin MA, Eryilmaz E, Karaisaoglu AO, Bukulmez A, Dogan I, Embleton DB, Cetinkursun S. Can mean platelet volume, Neutrophil-to-Lymphocyte, Lymphocyte-to-Monocyte, Platelet-to-Lymphocyte ratios be favourable predictors for the differential diagnosis of appendicitis? *J Pak Med Assoc*. 2019 May;69(5):647-654.
- Delgado-Miguel C, Muñoz-Serrano AJ, Núñez V, Estefanía K, Velayos M, Miguel-Ferrero M, Barrena S, Martínez L. Neutrophil-to-Lymphocyte Ratio as a Predictor of Postsurgical Intraabdominal Abscess in Children Operated for Acute Appendicitis. *Front Pediatr*. 2019 Oct 29;7:424. doi: 10.3389/fped.2019.00424.
- Güngör A, Göktuğ A, Güneylioğlu MM, Yaradılmış RM, Bodur I, Öztürk B, Karaman İ, Karacan CD, Tuygun N. Utility of biomarkers in predicting complicated appendicitis: can immature granulocyte percentage and C-reactive protein be used? *Postgrad Med*. 2021 Sep;133(7):817-821.
- Delgado-Miguel C, Muñoz-Serrano AJ, Barrena Delfa S, Núñez Cerezo V, Estefanía K, Velayos M, Serradilla J, Martínez Martínez L. Índice neutrófilo-linfocito como predictor de peritonitis en apendicitis aguda en niños [Neutrophil-to-lymphocyte ratio as a predictor of peritonitis in acute appendicitis in children]. *Cir Pediatr*. 2019 Oct 1;32(4):185-189.
- Greer D, Bennett P, Wagstaff B, Croaker D. Lymphopaenia in the diagnosis of paediatric appendicitis: a false sense of security? *ANZ J Surg*. 2019 Sep;89(9):1122-1125
- Celik B, Nalcacioglu H, Ozcatal M, Altuner Torun Y. Role of neutrophil-to-lymphocyte ratio and platelet-to-lymphocyte ratio in identifying complicated appendicitis in the pediatric emergency department. *Ulus Travma Acil Cerrahi Derg*. 2019 May;25(3):222-228. English.
- Virmani S, Prabhu PS, Sundeep PT, Kumar V. Role of laboratory markers in predicting severity of acute appendicitis. *Afr J Paediatr Surg*. 2018 Jan-Mar;15(1):1-4.
- Nazik S, Avci V, Küskü Kiraz Z. Ischemia-modified albumin and other inflammatory markers in the diagnosis of appendicitis in children. *Ulus Travma Acil Cerrahi Derg*. 2017 Jul;23(4):317-321.
- Yazici M, Ozkisacik S, Oztan MO, Gürsoy H. Neutrophil/lymphocyte ratio in the diagnosis of childhood appendicitis. *Turk J Pediatr*. 2010 Jul-Aug;52(4):400-3.

Consequently, I have one major suggestion about methodology:

- Please include NLR as an additional biomarker in the study and synthesis evidence on its diagnostic accuracy in comparison with the other included biomarkers. This would make the study novel, more relevant in the modern medicine, and different from what we already know about the diagnostic accuracy of commonly used biomarkers. This is a major revision which requires developing a new search strategy and subsequent data collection and analyses. Following this, the authors should add an extra section at the end

	of the methods section titled "Deviation from the registered protocol" to describe the reason for including an additional biomarker (In this case the reason being suggestion by a Reviewer).
--	---

REVIEWER	Reismann, Marc Charite Universitätsmedizin Berlin
REVIEW RETURNED	14-Nov-2021

GENERAL COMMENTS	At first glance the research question looks meaningful. In that sense the methodology seems to be appropriate and elaborated. However, the review is based on an insufficient definition of the inflammatory disease:.. appendicitis is not adequately defined at any stage. Thus, it is not clear which conditions are considered in the studies which have been included in the review. Further, important current evidence of acute appendicitis is not mentioned in the manuscript. The authors assume that acute appendicitis represents an uniform and necessarily progressive disease. Thus, they say that appendicitis is a necessarily "surgical condition" (p. 4). They "distinguish appendicitis from non-surgical case" (p.6). The way of looking at things is likely to neglect important aspects. Some of the most cited authors in the manuscript - Manne Andersson and Roland Andersson - hold the well-founded view that histopathologically phlegmonous and gangrenous appendicitis represent independent entities. According to these authors, phlegmonous appendicitis shows the strong tendency to resolve spontaneously. This view is even supported by current gene expression evidence (Kiss N et al., BJS Open 2021). Further, it has been shown that routine laboratory parameters not just differ significantly between these histopathological defined entities, but even substantially change over time within the equally affected patients (Minderjahn M et al., World J Ped, 2018; Reismann J et al., Ped Surg Int 2019) . These aspects should be implemented in the manuscript in some way. Either by a complete revision of the manuscript with differential analysis of the two entities, or at least by a substantial discussion why the authors differ from this view.
--

REVIEWER	Taylor, Kathryn University of Oxford, Nuffield Dept of Primary Care Health Sciences
REVIEW RETURNED	22-Feb-2022

GENERAL COMMENTS	The aim of this study is to summarise the evidence on the diagnostic accuracy of four blood tests for paediatric appendicitis by conducting a systematic review and meta-analysis. The study is well designed and the manuscript is well structured. Adherence to PRISMA guidelines is good and the stats are basically sound, but some aspects of the methods need to be clarified and the presentation could be improved in places. Methods Why were these 4 particular blood tests chosen for analysis? Three are stated as the "most widely "reported" in the literature i.e. the subject of studies. Which are most widely used in clinical settings?
---

	What is meant by the “optimal” reference standard? It needs a reference. The analysis of the 1000 hypothetical cohort could be described more clearly: (i) “reduction in patients requiring surgery” should read “reduction of patients having surgery” as “requiring surgery” are those who aren’t having surgery, according to the test, but need it i.e. false negatives. (ii) I would suggest stating the strategy of all 1000 hypothetical patients having surgery (without testing) first, and then the “reduction” in patients having surgery with the testing strategy would be clear. Supplementary figure 1 – Confidence intervals should be shown Supplementary figure 1 – WCC, NC and CRP has only 5 studies which are too few for meaningful analysis. It is not clear why it was not possible to include studies which did not report 2x2 data. The authors do not state what information was given instead (if any) for these studies. Perhaps the 2x2 data could have been estimated (from sensitivity, specificity and prevalence). Reporting the reasons for exclusion by study is only useful if the references are given. I don’t see the justification for removing outliers in sensitivity analysis. A study for which the diagnostic accuracy data were estimated would be grounds for exclusion in sensitivity analysis. Another example to exclude would be a study with primarily a paediatric population plus a few 18 year olds. Removing each study in turn is an approach to sensitivity analysis but just singling out an outlying study for exclusion when there is nothing questionable about it except its estimate being different to others. Differences in population could form subgroups and if there was a single study with a population that was very different to the others, its exclusion could feature in a sensitivity analysis. The authors refer to the optimal threshold. Why were they were not able to derive this using the approach described in ref 29 (Steinhauser 2016). The authors state as a study strength “Previous diagnostic accuracy meta-analysis methods have used diagnostic data for a single cut-off, and often the selected cut-off is the one maximizing the diagnostic accuracy parameters. Thus, by using our approach, we ensure the pooled diagnostic accuracy parameters are not overestimated²⁹” - It is not clear how the authors calculated the optimal threshold. Presentation The list of exclusions in methods is not easy to read. In “but a. data was not available in 2x2 format; b. a paediatric sub-group was not adequately detailed; c. exclusion....” the ‘a’ , ‘b’, etc need brackets i.e. (a), (b), etc. I would suggest that it might be useful to show the breakdown of n=60 by test in the PRISMA flow chart.
--	---

	I would suggest that “almost all” of 60 is 58 or 59, not 54 or less. It would be better to be more specific and rearranging the sentence to avoid starting with a number e.g. Of 60 studies, 54 (90%).. I appreciate that there are limits on the number of figures and tables that can be included with the main text and that this presents problems for this review as there are 4 blood tests and lots of studies for each. I would suggest that the authors are more selective in what they present in the main text and ensure that all main text figures are easy to read. The text in Figures 3 and 5 are illegible on a hard copy of this manuscript and when viewing the manuscript online the quality is poor. I don't see the need to have these figures in the main text. They just show heterogeneity. This point could be made in the main text and these figures could be split, enlarged and presented more clearly in the supplementary file. Or perhaps a single (enlarged) forest plot could be included in the main text, as illustration of the heterogeneity. I would also suggest moving Supplementary Table 6 to the main text. Currently, there is a lot of supplementary material and it is important that this material is clear. It can be improved in several ways: (a) The front page is very unhelpful as the tables have no names – “related to Table X” is a point that should be made in the footnote. (b) Separating the figure legends from the figures is not helpful. It's not clear why the supplementary figures are not included in the supplementary file (they could then be enlarged and would then be easier to read). (c) Placing the title of each table at the end of the table is unhelpful, particularly for tables that span over several pages. (d) The table of contents would also be improved by ensuring consistency of capitalisation and fonts.
--	--

REVIEWER	Hoq, Monsurul Murdoch Children's Research Institute, Clinical Epidemiology and Biostatistics Unit
REVIEW RETURNED	28-Feb-2022

GENERAL COMMENTS	The authors have provided adequate details of the statistical analysis done for the study. However, I have a minor comment regarding weighting. Did the author apply any weighting considering the sample size of the studies vary? I understand the in RevMan (Cochrane collaboration), Mantel-Haenszel is the default method, but would appreciate if this is clarified in the method. In the results section (page 11 2nd para), please report number along with percent. Please report median prevalence with inter quartile range. The findings from the sub-group analysis should be sign posted properly by highlighting the supplementary table 5 with a sub-heading. Otherwise its challenging to find estimates that support the results. The overall pooled estimates should be plotted in the forest plot along with the study specific estimates.
--

	Are these blood tests age-dependent? If not, the exclusion of age-dependent cut-offs from meta analysis could be justified. However, if the blood tests are age-dependents as reported by a few studies, a sub group analysis based on clinically important age-groups should have been done.
--	---

VERSION 1 – AUTHOR RESPONSE

Reviewer 1

4) To describe the performance of a diagnostic test, the positive and negative predictive values (PPV and NPV) are more relevant than sensitivity and specificity. Authors should added PPV (the chance of appendicitis if the test is positive) and NPV (the chance of non-appendicitis if the test is negative) in the table/figures.

- *Pooling predictive values is not recommended by the Cochrane as it is known that predictive values depend on prevalence which is likely to vary between studies. The Cochrane handbook argues that between study variation in prevalence may induce greater heterogeneity than sensitivity and specificity, and that the results will relate to the test at some average but unknown prevalence. Instead, to provide insight into the clinical interpretation of the pooled sensitivity and specificity we added the hypothetical cohort of 1000 children suspected of appendicitis to table 2.*

5) WCC, ANC, CRP and procal are actually non-specific inflammation markers. A positive blood test suggests the presence of inflammation but it would be difficult to differentiate appendicitis from other inflammatory conditions (such as bacteria gastroenteritis).Therefore, the above tests have a limited role in diagnosing appendicitis but can be used for ruling-out appendicitis.

- *This is a useful point, that we tried to consider in our conclusions. By aiming to only include studies where clinicians suspected appendicitis the inclusion criteria mimicked the clinical setting where conditions such as bacterial gastroenteritis may present similarly. One of the reasons for performing this review is to determine if these tests can differentiate appendicitis from other conditions. In tests such as CRP the specificity supports this. However, to reflect this comment we have stated in the discussion that there is greater value in any test for appendicitis in ruling out other conditions and blood tests alone should not be used for ruling in.*

6) As per what the authors described, the best performing blood tests for ruling-out pediatric appendicitis are WCC or ANC with sensitivities 0.87 and 0.90 respectively. However, more than 10 % of appendicitis would have been missed if WCC or ANC alone are used for diagnosis. A few papers have demonstrated that the combination of inflammatory markers has a dramatically improved diagnostic value for appendicitis [1–3]. The combined sensitivity of WCC, ANC and CRP is close to 100% [4]. In this study, the pooled sensitivity for combined WCC, ANC and CRP is only 80%, which could be due to the difference in interpreting combined test results.

- *The differences between the results is indeed due to the differences in interpreting combined test results. The test can be combined as all being positive (AND combination, increasing specificity) or all being negative (OR combination, increasing sensitivity). Therefore, we highlight this difference in the result section (**Supplementary table 5**). In addition, we present the results of both combinations using the individual patient data of 6 studies (**Table 2**). With the OR combination the sensitivity of CRP (10 mg/l) and WCC (10,000 cells/ul) is 97%, which is more in line with the result of the reviewer. In the updated literature search we also include the paper (Chiang et al) mentioned by the reviewer.*

7) The methods section did not specify what constitutes a positive combined test. Does a positive combined test require all 3 inflammation markers to exceed their cut-off values or any of the 3inflammation markers exceed their cut-off value? The sensitivity and NPV would only be raised if the positive combined test is defined as ANY of the inflammation markers exceed their cut-off value.

Chiang, et al found that WCC and ANC are more sensitive in early appendicitis, while the sensitivity of CRP is higher in late/complicated appendicitis. 99.7% of appendicitis has at least one of the inflammation markers raised [4]. In other words, if patients have normal WCC, ANC and CRP, the risk of appendicitis is extremely low (<0.3%). The author should relook into the studies for combined test and check if a negative combined test (all markers are normal) is a strong indicator for ruling-out appendicitis.

• *All studies that were included used an AND combination – i.e. all of the included tests had to be positive. In the updated literature search we included the study by Chiang et al which showed the results of the OR combination (i.e. all of the included test had to be negative). In addition, we have used IPD to report both AND combination of tests, as well as OR combinations to demonstrate the most specific and sensitive ways to use combined tests. We have also highlighted this by moving the results to the main table 2.*

8) The searching is not up to date (4th June 2019). Studies from the last 2 years should be included in this meta-analysis.

- *The search strategy has been updated in the new version of the manuscript to reflect the most recent studies. The new search date is 15th March 2022 and analysis has been updated to include new studies. This yielded more information with inclusion of 7 additional studies 7 reporting WCC, 4 on CRP, 3 on ANC, 3 on neutrophil percentage, one on procalcitonin and one on the combination of CRP and WCC.*

9) Small error: sup table 5, last two rows are duplicated

- *The last two rows of supplementary table 5 differ based on the cut off that was used for CRP in the combined tests of individual patient data (higher being a CRP cut off of 10mg/L, and lower being cut off of 50mg/L).*

Reviewer 2

10) Please include NLR as an additional biomarker in the study and synthesis evidence on its diagnostic accuracy in comparison with the other included biomarkers. This would make the study novel, more relevant in the modern medicine, and different from what we already know about the diagnostic accuracy of commonly used biomarkers. This is a major revision which requires developing a new search strategy and subsequent data collection and analyses. Following this, the authors should add an extra section at the end of the methods section titled “Deviation from the registered protocol” to describe the reason for including an additional biomarker (In this case the reason being suggestion by a Reviewer).

- *We appreciate that there are a number of studies that have recently been reporting NLR as a useful adjunct to the diagnosis of appendicitis. The reviewer outlines a number of papers to reflect this. However, a number of these papers were either considered or excluded during the repeat search strategy as they did not match inclusion criteria. Our search strategy did include terms for neutrophil and so we are confident that papers on NLR would not have been overlooked. Papers mentioned by the reviewer with reason for non-inclusion are listed below. In total our study identified 3 papers reporting NLR in the included review population. This was not enough to facilitate meta-analysis however we have mentioned it as a reported metric in the results section. :*
 - *Prasetya 2019 – excluded at title/abstract stage (compares appendicitis to intussusception).*
 - *Sengul 2020 – excluded at title/abstract stage (only includes operated children).*
 - *Tuncer 2019 – included in original manuscript, mention is made of the NLR in the results section.*
 - *Delgado-Miguel 2019 – excluded at title/abstract stage (only operated patients, compared appendix abscess with non-abscess).*
 - *Gungor 2021 – excluded at title/abstract stage (only includes operated patients)*

- *Delgado-Miguel 2019 – excluded at title/abstract stage (only includes operated patients)*
- *Greer 2019 – included in repeat search strategy, mention is made of the NLR in the results section*
- *Celik 2019 – excluded at title/abstract stage (only includes operated patients)*
- *Virmani 2018 – excluded at title/abstract stage (only includes operated patients)*
- *Nazik 2017 – excluded at full paper review stage (uses healthy controls)*
- *Yazici 2010 – included in original search strategy mention is made in the results section*

Reviewer 3

11) However, the review is based on an insufficient definition of the inflammatory disease: appendicitis is not adequately defined at any stage. Thus, it is not clear which conditions are considered in the studies which have been included in the review. Further, important current evidence of acute appendicitis is not mentioned in the manuscript. The authors assume that acute appendicitis represents a uniform and necessarily progressive disease. Thus, they say that appendicitis is a necessarily "surgical condition"(p. 4). They "distinguish appendicitis from non-surgical case" (p.6). The way of looking at things is likely to neglect important aspects. Some of the most cited authors in the manuscript - Manne Andersson and Roland Andersson - hold the well-founded view that histopathologically phlegmonous and gangrenous appendicitis represent independent entities. According to these authors, phlegmonous appendicitis shows the strong tendency to resolve spontaneously. This view is even supported by current gene expression evidence (Kiss N et al., BJSOpen 2021). Further, it has been shown that routine laboratory parameters not just differ significantly between these histopathological defined entities, but even substantially change over time within the equally affected patients (Minderjahn M et al., World J Ped, 2018;Reismann J et al., Ped Surg Int 2019) . These aspects should be implemented in the manuscript in some way. Either by a complete revision of the manuscript with differential analysis of the two entities, or at least by a substantial discussion why the authors differ from this view.

• *We appreciate that acute appendicitis does not represent one clinically distinct entity, and that children can present with a variety of severities and histopathological entities of appendicitis. This review aims to outline to value of ruling in or ruling out appendicitis with blood tests, and the inclusion criteria reflects each papers published classification of appendicitis and non-appendicitis patients. The broad nature of this means blood tests can be evaluated as a whole, and the sub-group analysis of histopathological types would be beyond the scope of this review and likely need a different case-control or cohort study. It would also require all patients to have undergone appendicectomy, and that would represent a different question to ours. To acknowledge the complexity of this we have added a reflection on this comment in the discussion (section – potential limitations).*

Reviewer 4

12) Why were these 4 particular blood tests chosen for analysis? Three are stated as the "most widely "reported" in the literature i.e. the subject of studies. Which are most widely used in clinical settings?

- *The blood tests outlined were based on these being the most widely reported in literature and used for different clinical scoring tools as outlined in the introduction. All are used widely in the clinical setting, with the exception of procalcitonin that was separated in the introduction to reflect this, but also included in results as a number of studies reported this. The search strategy was designed to include any tests reporting diagnostic accuracy for these, and would likely capture any commonly used other blood tests which would be compared to these benchmarks. If a large number of studies had reported a test not included in the design it would have been acknowledged. Some other included studies would report new diagnostic tests (e.g. Ozguner, I. F. et al. Are neutrophil CD64 expression and interleukin-6 early useful*

markers for diagnosis of acute appendicitis? *Eur. J. Pediatr. Surg.* 24, 179–183 (2014)) but as only reported in single studies comment was not made on these to avoid complicating the results.

13) What is meant by the “optimal” reference standard? It needs a reference.

- *Optimal reference standard was defined as part of the review to represent clinically what would confirm or refute a diagnosis of appendicitis. Histology to definitively confirm, and follow up of 14 days to refute. These standards were decided from clinical input and therefore not referenced. Additional reference standards were included to allow for studies that did not report histology (confirmed at the time of surgery) or may include non-operative appendicitis (none ended up having non-operative cases) to further reflect the methods by which clinicians diagnose appendicitis. Further detail has been added to justify the use of this optimal standard. Deviations from this optimal standard are reflected in the risk of bias (QUADAS-2) assessment, and this has been clarified also.*

14) The analysis of the 1000 hypothetical cohort could be described more clearly: (i) “reduction in patients requiring surgery” should read “reduction of patients having surgery” as “requiring surgery” are those who aren’t having surgery, according to the test, but need it i.e. false negatives.(ii) I would suggest stating the strategy of all 1000 hypothetical patients having surgery (without testing) first, and then the “reduction” in patients having surgery with the testing strategy would be clear.

- *We agree with this comment, this metric was used to reflect the effectiveness of a diagnostic test – the premise being if all patients were considered to have surgery, we wanted to illustrate how effective a test would be at reducing the intervention (surgery). On reflection this would not be a clinically appropriate use of these tests. The theoretical clinical application of the hypothetical cohort has been removed to help clarify this and we now only present the unnecessary treatment (false positive) and missed appendicitis (false negative) test results.*

15) Supplementary figure 1 – Confidence intervals should be shown

- *Confidence intervals have been added to the legend of this supplementary figure.*

16) Supplementary figure 1 – WCC, NC and CRP has only 5 studies which are too few for meaningful analysis.

- *We deleted this figure as we agree that this is not a meaningful analysis.*

17) It is not clear why it was not possible to include studies which did not report 2x2 data. The authors do not state what information was given instead (if any) for these studies. Perhaps the 2x2 data could have been estimated (from sensitivity, specificity and prevalence). Reporting the reasons for exclusion by study is only useful if the references are given.

- *There was a lack of clarity in this statement. Studies were included if 2x2 tables could be calculated from their reported sensitivity/specificity or any other metrics in the published papers. This has now been re-worded to reflect this in the section **Study selection and data extraction**. Reasons for exclusion of each study at the point of full paper review is also given in supplementary table 1 which is signposted in the main text.*

18) I don’t see the justification for removing outliers in sensitivity analysis. A study for which the diagnostic accuracy data were estimated would be grounds for exclusion in sensitivity analysis. Another example to exclude would be a study with primarily a paediatric population plus a few 18 year olds. Removing each study in turn is an approach to sensitivity analysis but just singling out an outlying study for exclusion when there is nothing questionable about it except its estimate being different to others. Differences in population could form subgroups and if there was a single study with a population that was very different to the others, its exclusion could feature in a sensitivity analysis.

- *We agree with the reviewer that there should be a reason for removing an outlier. Therefore, we removed this from our method section and result section.*

19) The authors refer to the optimal threshold. Why were they were not able to derive this using the approach described in ref 29 (Steinhauser 2016).

- *In the reference Steinhauser 2016, the authors applied the Youden index to identify the optimal threshold when sensitivity and specificity are both highest. In this setting we felt that this output would not be of benefit in clinical practice and instead focused on the cut-offs*

which generated the highest sensitivity (to rule out) and specificity (to rule in). Therefore we do not provide the optimal threshold, but provide insight into different commonly used cut offs which perform best for clinical application.

20) The authors state as a study strength “Previous diagnostic accuracy meta-analysis methods have used diagnostic data for a single cut-off, and often the selected cut-off is the one maximizing the diagnostic accuracy parameters. Thus, by using our approach, we ensure the pooled diagnostic accuracy parameters are not overestimated²⁹” - It is not clear how the authors calculated the optimal threshold.

- *The calculation of the optimal threshold is defined in the section **Data synthesis and statistical analysis**, where multiple cut-offs between studies were compared in the pooled analysis. The phrase optimal cut-offs in discussion reflected those with the highest sensitivity/specificity that was outlined in the results for each test as mentioned in response 19. To help clarify this further we have reworded that discussion point to say that we identified the best performing test parameter.*

21) The list of exclusions in methods is not easy to read. In “but a. data was not available in 2x2 format; b. a paediatric sub-group was not adequately detailed; c. exclusion....” the ‘a’, ‘b’, etc need brackets i.e. (a),(b), etc.

- *This paragraph has been reworded and the exclusion criteria made easier to read by the use of brackets.*

22) I would suggest that it might be useful to show the breakdown of n=60 by test in the PRISMA flow chart.

- *This information has been added to the PRISMA flow chart to add clarity.*

23) I would suggest that “almost all” of 60 is 58 or 59, not 54 or less. It would be better to be more specific and rearranging the sentence to avoid starting with a number e.g. Of 60 studies, 54 (90%)..

- *This has been rephrased and restructured to avoid the use of the phrasing “almost all”.*

24) I appreciate that there are limits on the number of figures and tables that can be included with the main text and that this presents problems for this review as there are 4 blood tests and lots of studies for each. I would suggest that the authors are more selective in what they present in the main text and ensure that all main text figures are easy to read. The text in Figures 3 and 5 are illegible on a hard copy of this manuscript and when viewing the manuscript online the quality is poor. I don’t see the need to have these figures in the main text. They just show heterogeneity. This point could be made in the main text and these figures could be split, enlarged and presented more clearly in the supplementary file. Or perhaps a single (enlarged) forest plot could be included in the main text, as illustration of the heterogeneity.

- *We agree that the plots for each test are useful for completeness, but due to the number of studies they are difficult to interpret. To respond to this feedback and aid presentation we have moved the plots to supplemental materials, where they could be enlarged and looked at in detail for those who wish to see full details of results by test.*

25) I would also suggest moving Supplementary Table 6 to the main text.

- *We agree that the impact of tests in a low prevalence is reduced by having a separate table of low prevalence (20%) in the supplementary materials. To address this and add clarity we have moved the metrics for low prevalence to main **table 2**. In addition we have looked further into the use of low prevalence in community practice and found recent evidence that 5% of cases in the community will have appendicitis (Blok G et al. Appendicitis in children with acute abdominal pain in primary care. Family practice, 758-65 (2021)) and so have updated the text to reflect this prevalence.*

26) Currently, there is a lot of supplementary material and it is important that this material is clear. It can be improved in several ways: (a) The front page is very unhelpful as the tables have no names – “related to Table X” is a point that should be made in the footnote.(b) Separating the figure legends from the figures is not helpful. It’s not clear why the supplementary figures are not included in the supplementary file (they could then be enlarged and would then be easier to read).(c) Placing the title of each table at the end of the table is unhelpful, particularly for tables that span over several

pages.(d) The table of contents would also be improved by ensuring consistency of capitalisation and fonts.

- *We have reworked the presentation of main figures as previously described, and the supplementary materials to help clarity. We have removed the “related to Table X” which was highlighted. Figures have been copied into the supplementary materials to respond to the issue of legends not being near figures. The titles have been placed at the beginning of each figure/ table.*

Reviewer 5

27) Did the author apply any weighting considering the sample size of the studies vary? I understand the in RevMan (Cochrane collaboration), Mantel-Haenszel is the default method, but would appreciate if this is clarified in the method.

- *To clarify, it is our understanding that Mantel-Haenszel is best applied for fixed effects for risk ratios or odds ratios. As this was a diagnostic accuracy meta-analysis we used bivariate random effects models. This is described in the methods section. “Meta-analysis was performed using a bivariate random effects model and pooled estimates of sensitivity, specificity, and likelihood ratios were generated when >4 studies per marker were included using the metandi module in STATA version 16”.*

28) In the results section (page 11 2nd para), please report number along with percent. Please report median prevalence with inter quartile range.

- *We have reported the number alongside percentage, and added the interquartile range to the reporting of median prevalence*

29) The findings from the sub-group analysis should be sign posted properly by highlighting the supplementary table 5 with a sub-heading. Otherwise its challenging to find estimates that support the results.

- *To add clarity to the use of sub-group / IPD analysis to assess these tests we have moved the results of this to the main **table 2**.*

30) The overall pooled estimates should be plotted in the forest plot along with the study specific estimates.

- *To address this we have provided pooled results in table 2 in the main text. All forest plot figures are moved to supplemental materials.*

31) Are these blood tests age-dependent? If not, the exclusion of age-dependent cut-offs from meta-analysis could be justified. However, if the blood tests are age-dependents as reported by a few studies, a subgroup analysis based on clinically important age-groups should have been done.

- *To our knowledge these tests are not age dependent in the paediatric population. For WCC, the studies (n=5) which reported differing cut offs for results based on patient age were too heterogeneous to perform sub-group analysis. For the neutrophils and ANC there were too few studies with age dependent cut offs to pool the results. As the majority utilized the same cut-off for included age groups, and in clinical practice results would not be differentiated by age it was decided a further sub-group analysis was not needed. To meet this point a comment about the discrepancy of age dependent cut offs is added to the discussion.*

VERSION 2 – REVIEW

REVIEWER	Yong, Chen KK women's and children's hospital , pediatric surgery
REVIEW RETURNED	07-Aug-2022
GENERAL COMMENTS	The paper reports a systematic review and meta-analysis of diagnostic accuracy of individual and combined serum inflammation markers for paediatric appendicitis based on 67

	studies Author concluded that the best performing blood tests for ruling-out pediatric appendicitis are WCC or ANC (sensitivity 0.85 and 0.90 respectively). The most specific tests were CRP alone ($\geq 50\text{mg/l}$, 38 studies, specificity 0.87 or CRP combined with WCC ($\geq 10,000$ cells/μl, six studies 0.93[95% CI 0.91-0.95]. Questions/comments for the authors: 1. In the conclusion , author described that the best tests to ruling out appendicitis are WBC or ANC as they only miss 40 and 60 of the 400 children with appendicitis. However, In table 2, author found that combined WBC or CRP (10mg/L) achieved a sensitivity of 97%. This combination would only miss 12 of 400 children with appendicitis and is much better than WBC and ANC alone. The important role of combined WBC or CRP is not highlighted in the manuscript. Author should added it in clinical impact and discussion. The conclusion should be change to : the best performing blood tests for ruling out paediatric appendicitis are combined WBC or CRP. 2. The combined markers (CRP and WBC) has completely different sensitivity/specificity and clinical implication compared to combine markers (CRP or WBC). Author analysed them separately in table 2 but failed to separate these 2 combinations in Supplementary figure 7 and 8. This resulted in a very confused and heterogeneous data. If author just presented combined WBC or CRP (huckins 2013, huckins 2016, Chiang 2020 and yap 2015), he would notice that all those studies has sensitivity more than 95%, suggesting the combined WBC or CRP are good markers for ruling out appendicitis.
--	--

REVIEWER	Taylor, Kathryn University of Oxford, Nuffield Dept of Primary Care Health Sciences
REVIEW RETURNED	09-Aug-2022

GENERAL COMMENTS	The revisions have met my concerns
------------------------------------

REVIEWER	Hoq, Monsurul Murdoch Children's Research Institute, Clinical Epidemiology and Biostatistics Unit
REVIEW RETURNED	17-Aug-2022

GENERAL COMMENTS	The authors have addressed the reviewer comments adequately. I have no further comments in terms of statistical analysis.
---

VERSION 2 – AUTHOR RESPONSE

Reviewer 1

1. In the conclusion , author described that the best tests to ruling out appendicitis are WBC or ANC as they only miss 40 and 60 of the 400 children with appendicitis. However, In table 2, author found that combined WBC or CRP (10mg/L) achieved a sensitivity of 97%. This combination would only

miss 12 of 400 children with appendicitis and is much better than WBC and ANC alone.

The important role of combined WBC or CRP is not highlighted in the manuscript. Author should added it in clinical impact and discussion. The conclusion should be change to : the best performing blood tests for ruling out paediatric appendicitis are combined WBC or CRP.

- We agree that the combination of tests can provide better metrics than single tests and this may have been understated when aiming to show the best single test. To increase the impact of this finding we have: highlighted the accuracy of combined WCC or CRP in the abstract results/conclusion; clarified this result in combined tests results; added an additional comment in the hypothetical cohort results; added this finding in the conclusion when discussing the most sensitive tests for ruling out. We hope this will balance the discussion of the best single tests and the finding of the superior accuracy of combined tests.

2. The combined markers (CRP and WBC) has completely different sensitivity/specificity and clinical implication compared to combine markers (CRP or WBC). Author analysed them separately in table 2 but failed to separate these 2 combinations in Supplementary figure 7 and 8. This resulted in a very confused and heterogeneous data. If author just presented combined WBC or CRP (huckins 2013, huckins 2016, Chiang 2020 and yap 2015), he would notice that all those studies has sensitivity more than 95%, suggesting the combined WBC or CRP are good markers for ruling out appendicitis.

- To add clarity to the comparison of AND/OR combinations we have amended Supplementary figure 7 and 8.

Reviewer 4

3. The revisions have met my concerns

- We thank the reviewer for reassessing our work and are glad that their concerns have been addressed.

Reviewer 5

4. The authors have addressed the reviewer comments adequately. I have no further comments in terms of statistical analysis.

- We thank the reviewer for reassessing our work and are glad their concerns have been addressed.